# Increased Abundances of CD16^+^ Non-Classical Monocytes Accompany with Elevated Monocytic PD-L1 and CD4^+^ T Cell Disturbances in Oropharyngeal Cancer

**DOI:** 10.3390/biomedicines10061363

**Published:** 2022-06-09

**Authors:** Christian Idel, Christina Polasky, Julika Ribbat-Idel, Kristin Loyal, Sven Perner, Dirk Rades, Karl-Ludwig Bruchhage, Ralph Pries

**Affiliations:** 1Department of Otorhinolaryngology, University of Luebeck, 23538 Luebeck, Germany; christian.idel@uksh.de (C.I.); christina.polasky@uksh.de (C.P.); kristin.loyal@uksh.de (K.L.); karl-ludwig.bruchhage@uksh.de (K.-L.B.); 2Department of Pathology, University of Luebeck, 23538 Luebeck, Germany; julika.ribbat-idel@uksh.de (J.R.-I.); sven.perner@uksh.de (S.P.); 3Department of Radiation Oncology, University of Luebeck, 23538 Luebeck, Germany; dirk.rades@uksh.de

**Keywords:** monocyte subsets, head and neck cancer, oropharyngeal cancer, HPV, PD-L1, T cells

## Abstract

Background: Patients with human papilloma virus (HPV)-related oropharyngeal cancer have a better prognosis than nonvirally associated patients, most likely because of better immune responses. Increased infiltration of T lymphocytes into the oropharyngeal tumor tissue has been observed, but the dynamics of circulating lymphocytes and monocytes are not fully understood. The aim of this study was to understand the population dynamics of circulating monocyte subsets in oropharyngeal cancer (OPC) patients with regard to the clinicopathological parameters and accompanying immunological consequences in view of the CD4/CD8 T cell subset composition, and the expression of checkpoint pathway proteins programmed death-1 (PD-1) and programmed death ligand-1 (PD-L1). Materials and Methods: The abundance of circulating monocyte subsets and peripheral blood CD4/CD8 T cells of oropharyngeal cancer patients and their PD-L1 and PD-1 expression levels were analyzed by flow cytometry. Results: The studied oropharyngeal cancer patients revealed heterogeneous individual redistributions of CD14^++^CD16^−^ (classical), CD14^++^CD16^+^ (intermediate), and CD14^dim+^CD16^+^ (nonclassical) monocyte subsets compared with healthy donors. These differences in monocyte subset alterations were independent in patients with TNM or HPV status but entailed further immunological consequences. Increased percentages of nonclassical monocytes significantly correlated with increased levels of monocytic PD-L1 expression. We observed significantly decreased levels of CD4^+^ effector T cells, which were accompanied by increased CD4^+^ effector memory T cells in OPC patients compared with healthy donors, each having a stronger effect in patients with decreased levels of classical monocytes. Conclusion: We conclude that oropharyngeal cancer, as a malignancy from a lymphoid-tissue-rich anatomical region, has a strong systemic impact on the differentiation and regulation of circulating innate and adaptive immune cells. Further comprehensive investigations are required for the possible future usability of the described immunological alterations as bioliquid parameters for prognosis or therapy response prediction.

## 1. Introduction

Head and neck squamous cell carcinoma (HNSCC) encompasses a heterogeneous set of distinct malignancies that develop from the mucosal epithelium in the oral cavity, pharynx, and larynx [1]. Different HNSCC subtypes exhibit different immunological behaviors, as well as different responses to therapeutic treatments [2,3]. Various clinical and biological features such as the stage, site of disease, comorbidities, or human papilloma virus (HPV) status contribute to the alteration of different immune cells within the malignant transformation process [4,5,6]. In this context, HPV is a well-established causative factor, especially for oropharyngeal cancer (OPC) as it is the most common HPV-related malignancy of the head and neck [7,8,9]. Patients suffering from HPV-related oropharyngeal cancer are mostly younger and have a more favorable prognosis than nonvirally associated oropharyngeal cancer patients [10,11]. Clinically, HPV-related OPCs are known to be poorly differentiated and often show a more positive lymph node status [12].

A better prognosis of HPV-related oropharyngeal cancer is most likely founded in better immune responses. Several studies, for instance, have shown an increased infiltration of CD8^+^ and CD4^+^ T lymphocytes within the oropharyngeal tumor tissue [13,14,15,16], but there are very few studies dealing with the dynamics of circulating lymphocytes and monocytes. In a recent study, a cohort of patients with different subtypes of head and neck squamous cell carcinoma (HNSCC) revealed lower overall proportions of intermediate monocytes and higher levels of classical monocytes compared with healthy donors [17]. Another study on 22 patients suffering from oropharyngeal cancer suggested an increased proportion of classical monocytes, but no difference in the proportion of nonclassical monocytes between OPC patients and healthy donors [18,19,20].

In general, monocytes can be classified into three subsets based on the gradual change in their CD14 (a lipopolysaccharide co-receptor) and CD16 (an immunoglobulin γ receptor) expressions [21,22,23]. CD14^++^CD16^−^ are called “classical” monocytes and comprise the biggest compartment, about 85% of the monocytes in healthy individuals. They are professional phagocytes that have the highest capacity to migrate and secrete the highest amounts of proinflammatory cytokines. CD14^+^CD16^+^ “intermediate” monocytes comprise about 5–10% of the major antigen presenting subtype. The remaining 5–10% constitute CD14^dim+^CD16^+^ nonclassical monocytes, which are supposed to be associated with anti-viral immunity due to their higher expression levels of Toll-like receptor 7 (TLR7) and TLR9 [24,25]. It has been hypothesized that both CD16^+^ subsets emerged from the classical monocyte population and are therefore more differentiated cells [26,27,28]. Monocytes are involved in human diseases through their functional effects, as well as through the differentiation potential of distinct subsets, whereas different diseases reveal shifts in different directions [29]. It was recently shown that recurrent respiratory papillomatosis (RRP) leads to an alteration in blood monocyte-derived immature Langerhans cells (iLCs) due to a reduced fraction of classical monocytes that generated most but not all of the iLCs [30]. RRP represents a rare disorder of HPV-driven premalignant airway lesions with a TH2-polarized adaptive immunity in papillomas and blood [31].

However, little is known about the peripheral monocyte composition in patients with oropharyngeal cancer, especially with regards to HPV-related and HPV-unrelated distributions. Therefore, the aim of this study was to understand the population dynamics of circulating monocyte subsets in oropharyngeal cancer patients with regard to clinicopathological parameters. Furthermore, a detailed analysis of the CD4/CD8 T cell subset composition, especially with respect to the immune checkpoint molecules programmed death-1 (PD-1)/programmed death ligand-1 (PD-L1) was addressed. The study aimed to increase our understanding of immunological changes in these functionally different monocyte subsets and their possible underlying immunological interplay in virally associated oropharyngeal cancer.

## 2. Materials and Methods

### 2.1. Ethics Statement

Peripheral blood samples were obtained from patients at the Department of Otorhinolaryngology, University Hospital Schleswig-Holstein, Campus Luebeck. The study was conducted in accordance with the WMA Declaration of Helsinki and approved by the ethics committee of the University of Luebeck (approval number 16-278).

### 2.2. Blood Collection and Patient Data

An informed written consent was signed by all blood donors. Blood samples were collected from healthy donors (*n* = 25; mean age of 58) and oropharyngeal cancer patients (*n* = 32; mean age of 65) by venipuncture into a sodium citrate S-Monovette (Sarstedt; Nümbrecht, Germany). The clinicopathological characteristics of the patients are listed in Table 1.

### 2.3. Profiling of Monocyte Subsets in Whole Blood

Within 4 h of blood collection, 20 µL of citrate blood was diluted in 80 µL PBS. Blood cells were stained with the following antibodies: CD45-PE, CD14-FITC, CD16-BV-510, HLA-DR-APC-Cy7, and CD3-PerCP (all from Biolegend, San Diego, CA USA). After 25 min of staining in the dark, 650 µL of red blood cell (RBC) lysis buffer (Biolegend) was added to the samples and incubated for another 20 min at room temperature. Subsequently, the suspension was centrifuged at 400× *g* for 5 min and supernatant was discarded. The cell pellet was gently resuspended in 100 µL of fresh PBS and used for flow cytometry.

### 2.4. Profiling of T Cell Subsets in Isolated PBMCs

Peripheral blood mononuclear cells (PBMCs) were isolated from the remaining blood by using Biocoll (Biochrom GmbH, Berlin/Heidelberg, Germany) in the density gradient centrifugation (400× *g* for 20 min). Next, the PBMC layer was transferred to a new 15 mL tube and washed once with PBS. The supernatant was discarded again and the PBMC pellet was resuspended in 1 mL PBS. For analysis of T cell subsets, 100 µL of the cell suspension was incubated with the following antibody cocktail: CD3-PerCP, CD4-PE-Cy7, CD8-BV-510, PD-1-PE, PD-L1-APC, CD45RA-APC-Cy7, and CCR7-BV421. The PD-L1 expression of monocytes was also analyzed during flow cytometry analysis of PBMCs. After 25 min of staining in the dark, the cells were washed with PBS and centrifuged at 400× *g* for 5 min. Cell pellets were resuspended in 200 µL PBS and used for flow cytometry.

### 2.5. Flow Cytometry Analysis

Flow cytometric analyses were carried out using a MACSQuant 10 flow cytometer (Miltenyi Biotec, Bergisch-Gladbach, Germany) in combination with FlowJo software version 10.0 (FlowJo, LLC, Ashland, TN, USA). All antibody titrations and proper compensation controls were performed beforehand using the same voltages as the samples. For whole-blood measurements, at least 100,000 CD45^+^ leukocytes were analyzed. The gating of monocyte subsets was performed as previously described (Polasky et al., 2021). For T cell analysis, 100,000 events within the PBMC gate were measured.

### 2.6. Statistical Analysis

GraphPad Prism Version 7.0f (GraphPad Software Inc, San Diego, CA, USA) was used for the statistical analyses conducted by testing for Gaussian distribution (normality tests), parametric (Student’s *t*-test), or nonparametric 1-way ANOVA with Bonferroni post hoc test. Correlation analyses were performed using the Pearson correlation coefficient: *p* < 0.05 (*), *p* < 0.01 (**), and *p* < 0.001 (***). Further statistical details are given in the figure legends, when required.

## 3. Results

### 3.1. Distribution of Circulating Monocyte Subsets in Oropharyngeal Cancer Patients

The gating strategy of CD14 and CD16 defined monocyte subsets was performed as published before [32]. CD45 was used as a pan leukocyte marker to facilitate whole-blood measurement, and monocytes were first roughly gated by their FSC/SSC characteristics and positivity for CD14 and CD16. Neutrophil granulocytes, NK cells, and B cells were excluded with the help of HLA-DR, which is specific for monocytes. The remaining real monocytes were then subgated into CD14^++^CD16^−^ (classical), CD14^++^CD16^+^ (intermediate), and CD14^dim+^CD16^+^ (nonclassical) monocytes.

Our investigation identified heterogeneous alterations in the abundance of all three monocyte subsets in oropharyngeal cancer patients compared healthy donors. OPC patients revealed similar overall median abundances of the monocyte subsets to the healthy donors, but much stronger dispersions of monocyte subset distributions (Figure 1A).

Next, based on their clinicopathological parameters, patients were subdivided into distinct groups with regard to their TNM and HPV status. No significant alterations in the monocyte subset composition were found in a comparison between patients with TNM1/TNM2 status and the cohort with TNM3/TNM4 status (Figure 1B). The subdivision of OPC patients with regard to their HPV status (p16^+^ vs. p16^−^) revealed no significant difference in the distribution of classical and nonclassical monocytes, but a significantly increased abundance of intermediate monocytes in p16^−^ negative oropharyngeal cancer patients (*p* = 0.0373; Figure 1C). Our data showed that oropharyngeal cancer patients revealed a wide spread distribution of increased, as well as decreased, levels of the three monocyte subsets compared with the situation of the healthy donors (Figure 1).

Thus, a question occurred about whether these different directions of monocyte subset distributions entail further immunological consequences. Therefore, based on the increased and decreased shift in the classical monocyte subsets, patients were subdivided into two distinct groups of more and less than the average value of 85% of peripheral classical monocytes (CMs) (Figure 2A).

As expected, this distribution was well-reflected in the abundance of the two CD16^+^ monocyte subsets in each cohort. OPC patients with decreased levels of classical monocytes exhibited highly significantly increased levels of both intermediate and nonclassical monocytes (Figure 2B). In addition, PD-L1 expression from pan-monocytes was analyzed from isolated PBMC. Our data revealed significantly increased monocytic PD-L1 expression levels of OPC patients with decreased CM levels compared with the increased-CMs cohort (*p* = 0.0099; Figure 2C).

### 3.2. Alteration of Circulating CD4^+^ T Cell Subsets in Oropharyngeal Cancer Patients

We also investigated the distribution of T-cell subsets in oropharyngeal cancer patients compared with healthy donors using flow cytometry. The T-cell differentiation from naïve to effector, effector memory, and central memory cells was analyzed for CD4^+^ and CD8^+^ T cells by specific markers. Therefore, PBMCs were first roughly gated by FSC/SCC characteristics and any doublets were excluded. The CD3^+^ population represents T cells, which were further subgated into CD4 or CD8 subsets.

The T-cell differentiation state was classified with the help of CD45RA and CCR7 into naïve (CD45RA^+^ CCR7^+^), central memory (CD45RA^−^ CCR7^+^), effector memory (CD45RA^−^ CCR7^−^), and effector (CD45RA^+^ CCR7^−^) T cells for CD4^+^ and CD8^+^. Percentages of PD-1-expressing cells of both CD4 and CD8 T lymphocytes were not significantly altered in oropharyngeal cancer patients compared with healthy donors (Figure 3).

We observed significantly decreased levels of CD4^+^ effector T cells accompanied by an increase in CD4^+^ effector memory T cells in OPC patients compared with healthy donors, each with stronger effects in patients with decreased levels of classical monocytes (Figure 4). No significant differences were found in naïve or central memory CD4^+^ T cells and CD8^+^ T cell subsets, respectively (Figure 4).

### 3.3. Alteration of Monocyte Subsets and Accompanying Consequences

Our data revealed different kinds of monocyte subset alterations in patients with oropharyngeal cancer with different immunological consequences in view of monocytic PD-L1 expression and altered percentages of CD4^+^ effector and CD4^+^ effector memory T cells. The correlation analyses of these parameters revealed correlations by tendency between classical monocyte percentages and monocytic PD-L1 expression, as well as between classical monocyte percentages and percentages of CD4^+^ effector memory T cells (Figure 5A,B).

Two different CD16^+^ monocyte subsets, intermediate and nonclassical, can emerge from the classical monocyte population and were therefore considered individually. Increased percentages of non-classical-monocytes, but not intermediate monocytes, significantly correlated with the levels of monocytic PD-L1 expression and correlated by tendency with the percentage of CD4^+^ effector memory T cells (Figure 5).

## 4. Discussion

The aim of this study was to investigate the influence of oropharyngeal cancer (OPC) on the abundance of peripheral blood monocyte subsets and immune alterations in CD4/CD8 T cell subsets. Furthermore, the expression of PD-1/PD-L1 in OPC patients and healthy donors was analyzed for CD3^+^CD4^+^ and CD3^+^CD8^+^ subsets, and monocytes using flow cytometry.

Under healthy conditions, around 85% of monocytes are classical CD14^+^CD16^−^ monocytes, which are characterized by phagocytosis, endothelial transmigration, and production of proinflammatory cytokines. The remaining 15% are linked to more specialized antigen presentation functions, defense against viruses, and patrolling behavior, whereas different human diseases result in different displacements of the monocyte subset distribution [29]. Under various inflammatory conditions, increased levels of CD16^+^ intermediate and nonclassical subsets have been reported. These subsets are generally termed proinflammatory because of their high production of TNF-α and IL-1β [21]. Elevated abundances of different antigen-presenting cells such as myeloid dendritic cells (mDCs), plasmacytoid dendritic cells (pDCs), or monocytes was shown in HPV-related head and neck cancer [33]. We recently showed that obstructive sleep apnea syndrome (OSAS) patients reveal a redistribution of monocyte subsets and a subsequently affected imbalance in the PD-1/PD-L1-mediated communication with CD4/CD8 T cells. Decreased percentages of CD14^++^CD16^−^ classical monocytes correlated with increased levels of intermediate and nonclassical monocytes [32]. Similar alterations of decreased classical monocyte subset percentages have also been shown in asthma patients [34].

Examinations of the chronic skin disease, rosacea, revealed increased frequencies of classical monocytes, but not intermediate or nonclassical monocytes [35]. The underlying molecular mechanisms of these different cellular shifts are still largely not understood. Korenfeld et al. identified selective deficiencies in peripheral circulating nonclassical and intermediate monocyte subsets, and significantly increased percentages of classical monocytes in patients with multiorgan autoimmunity, lymphoproliferation, and recurrent infections, respectively. They found activating mutations of transcription factor STAT3 in these patients, which suggests a modulated regulation of the hematopoietic cell differentiation [36]. The present study revealed distinct redistributions of monocyte subsets in the peripheral blood of OPC patients, which show increased levels of classical monocytes in some patients as well as decreased CM levels in others, which are not significantly correlated with patients with HPV or TNM. We therefore asked whether these individual conditions of increased and decreased abundances of classical monocytes within the cohort of oropharyngeal cancer patients are paired with additional immunologic alterations.

OPC patients with decreased abundances of classical monocytes revealed significantly increased levels of monocytic PD-L1 compared with the cohort of increased abundances of classical monocytes. PD-L1 is a checkpoint immunoregulatory molecule, and is involved in tumor escape mechanisms from T-cell immune responses [37]. Different myeloid cells such as monocytes, macrophages, and myeloid-derived suppressor cells (MDSCs) are known to express PD-L1 in a tumor microenvironment, thereby suppressing T-cell functions [38,39]. Furthermore, expression of PD-L1 on myeloid cells has been correlated with poor prognosis in different tumor entities [40,41,42]. These data suggest a higher level of immune suppression in oropharyngeal cancer patients with increased abundances of CD16^+^ monocytes.

Most analyzed T-cell subsets were not significantly different between OPC patients and healthy donors. Significantly decreased percentages of CD4^+^ effector cells as well as increased levels of CD4^+^ effector memory cells were found in our study in OPC patients, each with a greater shift in patients with decreased levels of nonclassical monocytes.

Memory T cells have the potential for long-term survival, while effector T cells are short-lived cells. In healthy individuals, memory T cells are quickly converted into large numbers of effector T cells upon activation [43]. A similar shift from naïve to effector memory T cells was noted in patients with HPV-related oropharynx carcinomas, but they also found that a number of HPV-negative patients had high percentages of effector memory T cells [44]. Thus, this may primarily be due to the fact that oropharyngeal carcinoma arises from tonsils, which consist of lymphoepithelial tissue, and therefore may show more efficient immune responses than tumors from other sites, independent of individual HPV status [45]. Wansom et al. observed that numbers of tumor-infiltrating lymphocytes (TILs) correlated with a better prognosis. A good correlation was also found between disease-specific survival and HPV status. However, they did not find significantly more TILs in HPV positive patients [46]. Here, we documented that OPC patients with decreased percentages of classical monocytes and increased levels of nonclassical monocytes revealed significantly increased percentages of effector memory CD4^+^ T cells, regardless of HPV status, which suggests a higher level of intrinsic immunogenicity in these patients. In patients with rheumatoid arthritis, it was observed that circulating CD16^+^ monocyte subsets revealed the potential to promote Th17 cell expansion from peripheral blood memory CD4^+^ T cells in vitro [47]. Whether this is also the case in patients with head and neck cancer needs to be examined in future investigations.

An acknowledged limitation of the present study is the relatively small number of patients in the analysis of monocyte subsets. Further comprehensive investigations on bigger cohorts of oropharyngeal cancer patients and additional checkpoint regulatory molecules are required to elucidate the interplay of innate and adaptive immune cell subsets in the peripheral blood of patients suffering from this disease. In addition, survival analysis over a longer period is also necessary to elucidate the possible future usability of the described immunological alterations for prognosis or therapy response prediction.

## Figures and Tables

**Figure 1 biomedicines-10-01363-f001:**
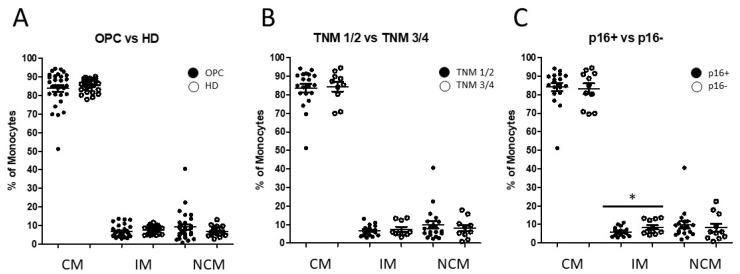
Flow cytometric analysis of CD14- and CD16-characterized monocyte subsets. (**A**) Whole-blood analysis revealed similar median abundances of classical monocytes (CMs), intermediate monocytes (IMs), and nonclassical monocytes (NCMs) in oropharyngeal cancer patients (*n* = 32) compared with healthy donors (*n* = 25), but much stronger dispersions of monocyte subset distributions in cancer patients. (**B**) No significant alterations in the monocyte subset composition were found between patients with TNM1/2 and the cohort with TNM3/4. (**C**) Subdivision with regard to the HPV status (p16^+^ vs. p16^−^) revealed no significant differences in classical and nonclassical monocyte compositions, but slightly significantly increased levels of intermediate monocytes in p16^−^ negative oropharyngeal cancer patients. * *p* < 0.05.

**Figure 2 biomedicines-10-01363-f002:**
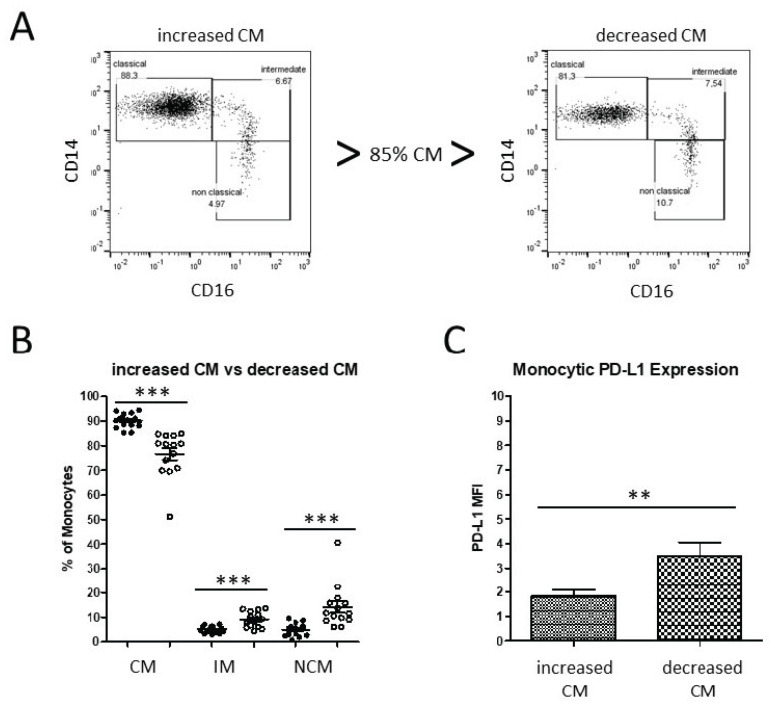
Flow cytometric analysis of two distinct OPC patient cohorts. (**A**) Analyses of two representative patients with increased and decreased abundances of peripheral classical monocytes. (**B**) OPC patients with decreased classical monocytes revealed significantly increased levels of both intermediate and nonclassical monocytes. (**C**) OPC patients with decreased classical monocytes revealed significantly increased expression levels of monocytic PD-L1. ** *p* < 0.01; *** *p* < 0.001.

**Figure 3 biomedicines-10-01363-f003:**
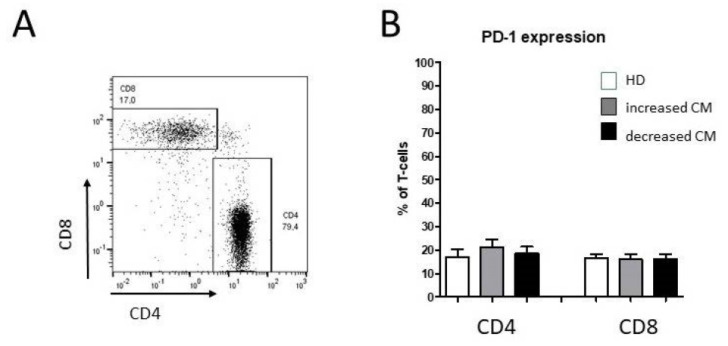
Flow cytometric analysis of CD4^+^ and CD8^+^ T cells in PBMCs from healthy donors and oropharyngeal cancer patients. (**A**) Example gating scheme of CD4^+^ and CD8^+^ T cells from one representative patient. (**B**) Percentages of PD-1^+^ cells within both the CD4^+^ and CD8^+^ T -cell subset.

**Figure 4 biomedicines-10-01363-f004:**
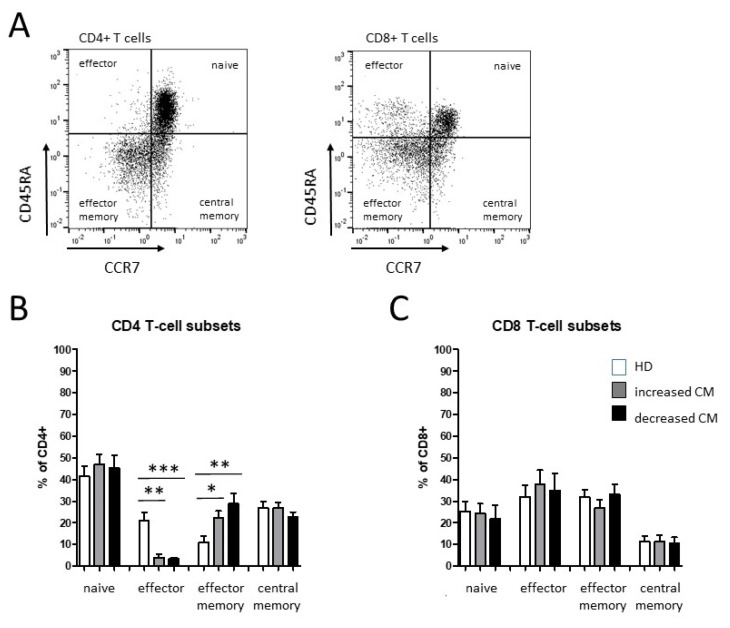
Flow cytometric analysis of CD4^+^ and CD8^+^ T-cell subset distribution in PBMCs from healthy donors and oropharyngeal cancer patients. (**A**) Example gating scheme of CD4^+^ and CD8^+^ T-cell subsets from one representative patient. Based on expressions of CD45RA and CCR7, naïve (CD45RA^+^ CCR7^+^), central memory (CD45RA^−^ CCR7^+^), effector memory (CD45RA^−^ CCR7^−^), and effector (CD45RA^+^ CCR7^−^) T cells were identified. (**B**) Percentages of naïve, effector, effector memory, and central memory T cells within CD4^+^ and CD8^+^ T cells (**C**) from healthy donors and OPC patients with increased and decreased levels of classical monocytes (CM). * *p* < 0.05; ** *p* < 0.01; *** *p* < 0.001.

**Figure 5 biomedicines-10-01363-f005:**
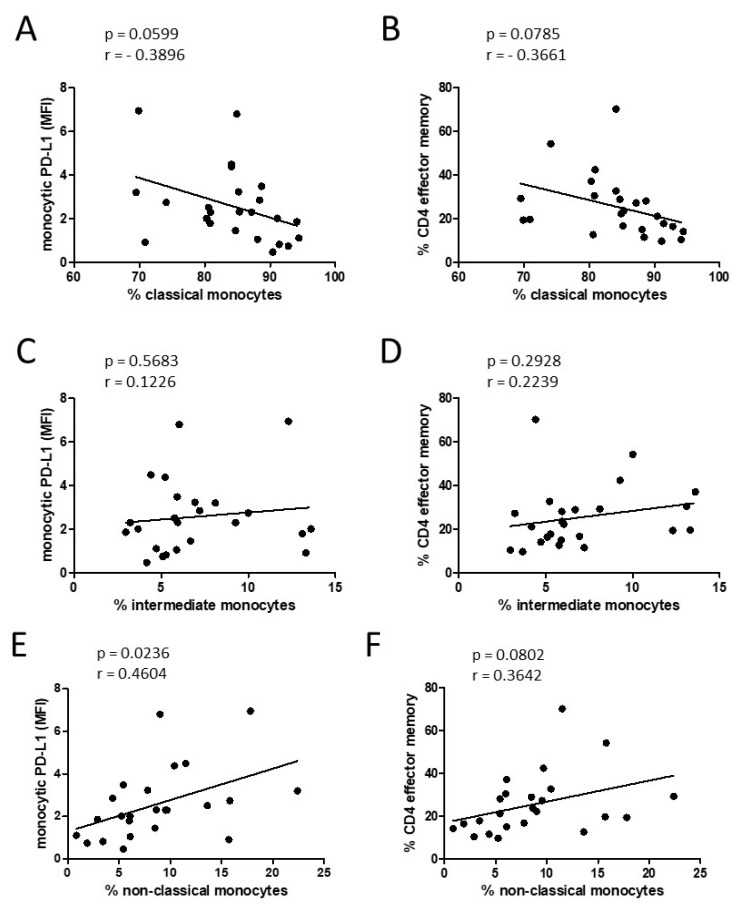
Correlation analysis between different parameters measured in oropharyngeal cancer patients. Abundances of classical monocytes (**A**,**B**), intermediate monocytes (**C**,**D**), and nonclassical monocytes (**E**,**F**) correlated with mean fluorescence intensities (MFIs) of monocytic PD-L1 and percentages of CD4 effector memory T cells. A multivariate progression with the Pearson correlation was performed. The correlation coefficient (r) and *p* values are given for each pair. *p* < 0.05 was considered significant.

**Table 1 biomedicines-10-01363-t001:** Clinicopathological parameters.

Characteristics	Patients (*n* = 32)	
	*n*	%
Sex		
m	26	81
f	6	19
**Age (years)**		
≤ 60	9	28
> 60	23	72
**Tumor Stage**		
T1–T2	21	66
T3–T4	11	34
**Nodal Stage**		
N0	6	19
N1	11	34
N2	10	31
N3	5	16
**Distant Metastasis**		
M0	32	100
**HPV status**		
Positive	21	66
Negative	11	34
**UICC stage**		
I	1	3
II	4	13
III	7	22
Obsolete	20	63
**Alcohol abuse**		
Yes	6	19
No	26	81
**Tobacco consumption**		
Yes	24	75
No	8	25

## Data Availability

The data presented in this study are available on request from the corresponding author.

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
