# Peer review of "Increased Abundances of CD16+ Non-Classical Monocytes Accompany with Elevated Monocytic PD-L1 and CD4+ T Cell Disturbances in Oropharyngeal Cancer"

_biomedicines, 2022, doi:10.3390/biomedicines10061363_

Round 1
Reviewer 1 Report
The authors wanted to investigate the connection between HPV on oropharyngeal cancer (OPC). They believe that a better prognosis from HPV-associated OPC results from better immune responses. There is better infiltration of CD8+ and CD4+ T lymphocytes in these cases. The authors wish to evaluate the circulating T lymphocytes, which have a significant lack of knowledge. Under healthy conditions, the dominant form is classical CD14+CD16- monocytes, while under inflammatory conditions, there are increased levels of CD16+. The authors wished to investigate the heterogeneous three categories of monocytes CD14+CD16- (classical), CD14+CD16+ (intermediate), and CD14dimCD16+ (non-classical) monocytes. Their investigation found significant heterogeneous alterations in OPC patients compared to healthy donors. Furthermore, the authors investigated PD-L1, which is important in suppressing the T-cell immune response and CD4+ effector memory cells. They found a strong correlation between classical and non-classical populations in both cases.
The authors conducted a thorough investigation of the heterogeneity of monocytes within their patients and conducted extensive background information before evaluating the monocyte populations. Furthermore, the data represented follows a significant trend even with such a small population of patients. In most cases, I think the author's data show significant support for their hypothesis.
One weakness of the manuscript is the flow parameters specified that at least 100,000 cells were quarried. I was hoping the authors could comment on instances where more than 100,000 were tested and if they observed any changes in the population distribution. In addition, the author's hypothesis could further be strengthened by providing a comment on the population's background and whether there was a correlation with parameters like gender, age, or tobacco use. In some of these cases, a significant bias might need to be addressed.
Overall I think this is a good manuscript with solid support for the author's hypothesis. Furthermore, the correlation of the immune and inflammatory responses to cancer progression is a vital area of research that still warrants further investigation.
Author Response
Dear Reviewer, thank you for your kind comments.
We will keep in mind your comments in terms of potential correlations with additional clinical parameters and aim to analyze it in a bigger cohort.
Reviewer 2 Report
I do not have any particular comments. The authors pointed out well the limits of the study, that the patient number is quite low, thus should be extended, however, the results are interesting and significant enough to be published in the biomedicines.
Author Response
Dear Reviewer, thank you very much for your kind comments.